# Solar-driven membrane separation for direct lithium extraction from artificial salt-lake brine

Shenxiang Zhang[1,2], Xian Wei[1,2], Xue Cao[1,2], Meiwen Peng[3], Min Wang[4], Lin Jiang [3] ✉ & Jian Jin [1,2] ✉

The demand for lithium extraction from salt-lake brines is increasing to address the lithium supply shortage. Nanofiltration separation technology with high $Mg^{2+}/Li^+$ separation efficiency has shown great potential for lithium extraction. However, it usually requires diluting the brine with a large quantity of freshwater and only yields $Li^+$-enriched solution. Inspired by the process of selective ion uptake and salt secretion in mangroves, we report here the direct extraction of lithium from salt-lake brines by utilizing the synergistic effect of ion separation membrane and solar-driven evaporator. The ion separation membrane-based solar evaporator is a multilayer structure consisting of an upper photothermal layer to evaporate water, a hydrophilic porous membrane in the middle to generate capillary pressure as the driving force for water transport, and an ultrathin ion separation membrane at the bottom to allow $Li^+$ to pass through and block other multivalent ions. This process exhibits excellent lithium extraction capability. When treating artificial salt-lake brine with salt concentration as high as 348.4 g $L^{-1}$, the $Mg^{2+}/Li^+$ ratio is reduced by 66 times (from 19.8 to 0.3). This research combines ion separation with solar-driven evaporation to directly obtain LiCl powder, providing an efficient and sustainable approach for lithium extraction.

Lithium, one of the most valuable resources, has found its way into various industries, ranging from ceramics, glass, pharmaceuticals, and nuclear to the booming lithium battery technology[1-4]. The rapid growth in lithium consumption, spurred by the expansion of the lithium battery market in recent years, has made it crucial to source lithium from various channels[5-10]. Lithium sourcing from salt-lake brines accounts for ≈70% of recoverable lithium on land[11]. It has become a vital supply route to ensure the healthy development of the lithium battery market. However, lithium extraction from salt-lake brines is challenging because of the high concentration of competing

ions, like $Mg^{2+}$. Compared to conventional ion separation technologies such as ion exchange[12], electrodialysis[13], and solvent extraction[14], nanofiltration (NF) membrane separation[15-17] is considered one of the most efficient methods for extracting lithium from brine. The combination of size sieving and Donnan exclusion in NF membranes offers an ideal option for achieving effective monovalent/divalent ion separation, particularly for $Li^+/Mg^{2+}$ separation[18-22]. To enhance the performance of $Li^+$ separation, various investigations have been carried out, including developing innovative NF membrane materials and optimizing the membrane structure and surface properties[23-26]. Despite

[1]College of Chemistry, Chemical Engineering and Materials Science, Jiangsu Key Laboratory of Advanced Functional Polymer Design and Application, Soochow University, Suzhou, Jiangsu, China. [2]Collaborative Innovation Center of Suzhou Nano Science and Technology, Jiangsu Key Laboratory of Advanced Negative Carbon Technologies, Soochow University, Suzhou, Jiangsu, China. [3]Institute of Functional Nano & Soft Materials (FUNSOM), Jiangsu Key Laboratory for Carbon-Based Functional Materials & Devices, Soochow University, Suzhou, Jiangsu, China. [4]Key Laboratory of Comprehensive and Highly Efficient Utilization of Salt Lake Resources, Qinghai Institute of Salt Lakes, Chinese Academy of Sciences, Xining, Qinghai, China. ✉e-mail: ljiang@suda.edu.cn; jjin@suda.edu.cn

significant advancements in the separation of $Mg^{2+}$ and $Li^+$ using NF membrane processes, highly concentrated brines generate a high osmotic pressure difference across the NF membrane, requiring high applied pressure that may exceed the mechanical strength of the membrane module. Therefore, the conventional NF process typically involves the 10−20 times dilution of high-concentration brine with a large quantity of freshwater in the pre-treatment stage[14,27,28]. Moreover, NF membrane separation processes solely yield $Li^+$-enriched solutions, necessitating further concentration to obtain solid lithium products, which requires energy-intensive processes like thermal distillation or high-pressure reverse osmosis to obtain the $Li^+$-enriched solution with the required concentration. Hence, innovative material and separation process design is required to achieve more efficient and cost-effective Mg/Li separation.

Interfacial solar evaporation, a sustainable and cost-effective technology for clean water production, harnesses solar energy and concentrates generated heat to facilitate water evaporation[29–36]. Additionally, this approach offers a promising avenue for salt resource recovery from seawater and brines[37–40]. During evaporation, salt ions become concentrated and eventually precipitate out. Selective ion uptake and transport driven by interfacial solar evaporation are common processes in plants. For instance, mangrove is a type of salt-tolerant tree that thrives in tropical and subtropical coastal regions (Fig. 1a)[41,42]. Mangroves can regulate the uptake and release of ions to adapt to the saline or brackish environment[43]. When provided with optimal sunlight and temperature, the stomata in leaves open, causing water molecules to be released into the atmosphere as vapor. This process is followed by water flow towards the stomata, effectively replacing the evaporated water molecules while ensuring the continuous upward flow of water and ions toward the leaves[44,45]. To survive in seawater, ion pumps and channels in cell membranes selectively transport ions through mangrove roots (Fig. 1b)[46,47]. Additionally, salt glands located in leaves enable mangroves to directly secrete salt out of their systems (inset in Fig. 1a), further aiding their ability to survive in saline environments[48,49].

Inspired by the process of selective ion uptake and salt secretion in mangroves, we utilize the synergistic effect of ion separation membrane and solar-driven evaporator to direct extraction of lithium chloride (LiCl) powder from salt-lake brines. Figure 1c illustrates the structure of the membrane-based solar-driven evaporator, composed of three functional layers. To realize a high photothermal conversion efficiency, vertically aligned polyaniline (PANI) nanofiber array was selected as the upper photothermal layer owing to its sunlight absorption capacity. In the middle, a hydrophilic poly(ether sulfone) (PES) macroporous membrane with interconnected water-conducting channels generates capillary pressures serving as the driving force for water transport. At the bottom is an ultrathin polyamide (PA) ion separation membrane allowing $Li^+$ to pass through and block other multivalent ions, like $Ca^{2+}$ and $Mg^{2+}$. When treating a $LiCl/MgCl_2$ mixed solution, water and $Li^+$ ions trans through the PA membrane to reach the photothermal layer under capillary pressure. As water continually evaporates, solid LiCl powders can be collected directly on the surface of the evaporator. This ion separation membrane-based solar evaporator maintains a stable lithium enrichment even when processing high-concentration salt solutions. This study demonstrates the amazing capability of lithium enrichment. It provides an effective way to directly obtain high-purity lithium from salt-lake brines only using sustainable energy.

## Results and discussion
### Fabrication and characterization of PANI nanoarrays solar evaporator

Figure 2a demonstrates the schematic illustration of the PANI nanoarrays solar evaporator. The PANI nanoarrays were grown in situ

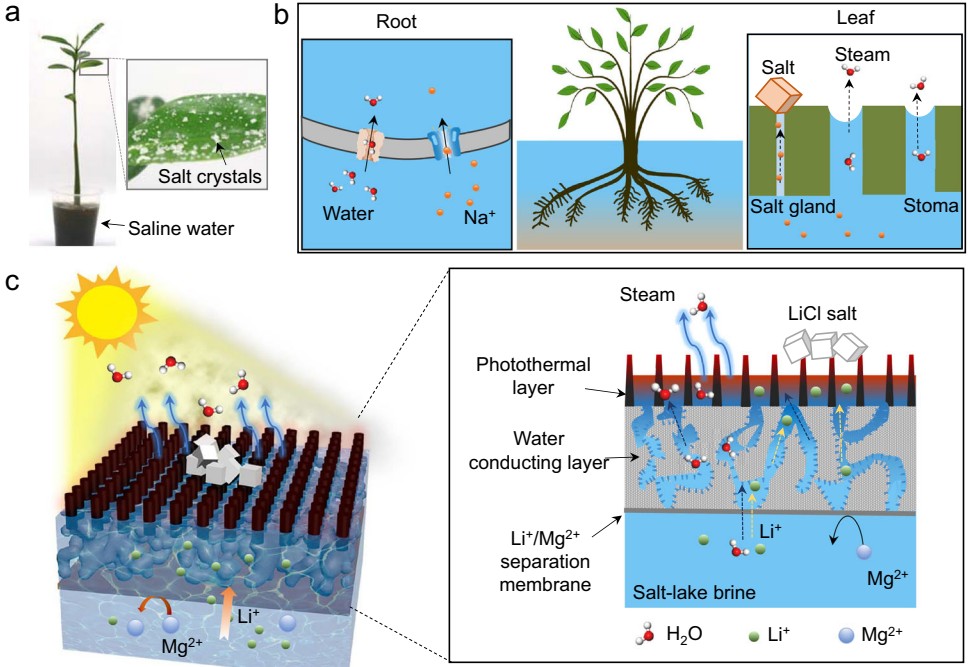

**Fig. 1 | Design of solar-driven membrane separation for lithium extraction.**
**a** Photograph of mangrove and leaf blade with salt crystals on the leaf surface. To observe the salt secretion phenomenon, this mangrove was placed in a 50 g/L NaCl solution for 4 weeks. **b** A schematic of regulating water and ion intake and release in mangroves. The root has transport proteins that selectively transport water and specific ions across the cell membranes. The leaf absorbs solar energy and uses it to drive water's evaporation through the stomata. Also, the salt glands in leaves enable mangroves to secrete excess salt out of leaves directly. **c** Working principle of polyamide membrane-based solar-driven lithium extraction. The photothermal layer absorbs solar energy to heat water, causing water evaporation. The water-conducting layer generates capillary force, encouraging a continuous upward flow of water and ions from the bottom to the surface. The ion-selective membrane allows for rapid diffusion of $Li^+$ while rejecting $Mg^{2+}$. As water continually evaporates, solid LiCl powder can be collected on the surface of the solar evaporator.

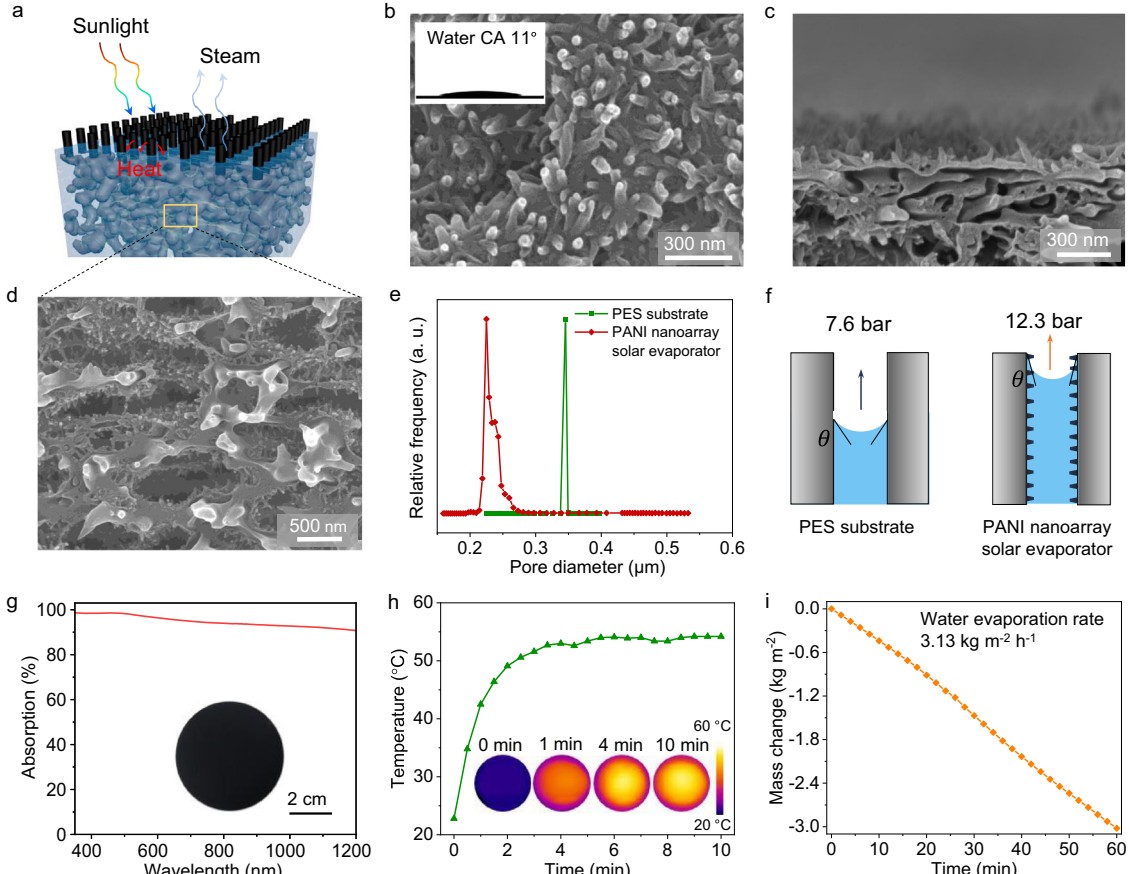

**Fig. 2 | Characterization of PANI nanoarrays solar evaporator. a** Schematic illustration of PANI nanoarrays solar evaporator. **b** Surface SEM image of PANI nanoarrays evaporator. Inset is the instantaneous CA of a water drop on the PANI nanoarrays evaporator surface. Cross-sectional SEM images of PANI nanoarrays evaporator with an enlarged view of **c** the top layer and **d** the inner section. **e** Pore size distribution of PES substrate and PANI nanoarrays evaporator. **f** Schematic description of capillary pressure generated in PES substrate and PANI nanoarrays evaporator. **g** Light absorption spectrum of PANI nanoarrays evaporator. Inset is a photograph of the PANI nanoarrays evaporator. **h** Time-dependent surface temperature of wet PANI nanoarrays evaporator, under 3 sun illumination. Insets are infrared thermal images of PANI nanoarrays evaporator under 3 sun illumination. **i** Time-dependent mass change of water due to the evaporation through PANI nanoarrays solar evaporator under 3 sun illumination. Source data are provided as a Source Data file.

on a PES macroporous membrane with a sponge-like pore structure (Supplementary Fig. 1) that serves as a substrate. Nanofibers were formed through a polymerization process of aniline monomers using ammonium persulfate[50,51]. Scanning electron microscope (SEM) images in Fig. 2b–d shows that the surface and internal channels of the PES substrate were coated by vertically aligned PANI nanofibers with 30–50 nm in diameter and 50–250 nm in length. The resulting PANI nanoarrays-coated PES membrane was hydrophilic, with an instantaneous water contact angle (CA) of 11° (inset in Fig. 2b). The pore size distribution of the PES substrate before and after the growth of PANI nanoarrays was measured using the bubble pressure method. As demonstrated in Fig. 2e, the average pore diameter of the PES substrate reduced from 345 nm to 225 nm due to the growth of PANI nanoarrays on its inner surface.

A stable and efficient water supply is a crucial component of the solar evaporator. During the steady interface evaporation process, water molecules removed by evaporation are replaced by water flow toward the air-water interface. The cohesive force between water molecules is crucial in maintaining a continuous water supply, generating tension between the molecules[44,45]. The water surface tension that forms in micrometer-sized pores or channels can generate a capillary pressure that is the primary driving force for water transport in a solar evaporator. The capillary pressure ($P_c$) can be estimated using the Young-Laplace equation, $P_c = 4\gamma cos\theta/d$, where $\gamma$ is the surface tension of water, $\theta$ is the water CA on the pore surface, and $d$ is the pore

diameter. For the above equation, $\gamma$ is 0.071 N m$^{-1}$ at 30 °C, assuming $\theta$ is the water CA on the membrane surface. The pores or channels in PES substrate and PANI nanoarrays solar evaporator can produce capillary pressures of 7.6 and 12.3 bar in the liquid phase, respectively (Fig. 2f). This result proves that smaller pore size and a hydrophilic surface can increase capillary pressure.

Efficiently absorbing broad-spectrum sunlight and converting it into heat are important steps in solar-driven evaporation. The PANI evaporator presented a dark surface resulting from its exceptional light absorption capacity, particularly in the visible range, reaching as high as 96% (Fig. 2g). The PANI nanoarrays efficiently trap incident light and promote its multiple reflectance until absorption, contributing to the high absorption capacity[52]. To evaluate the light-to-heat conversion performance, the surface temperature of the PANI nanoarrays evaporator was monitored under 3 sun illumination using an infrared thermal imager (Fig. 2h). Within 2 min of illumination, the surface temperature of the wet PANI nanoarrays evaporator increased rapidly from 22.0 to 49.2 °C. After 5 min, the surface temperature reached a maximum of 54.0 °C and remained stable. The light-to-heat conversion performance can be attributed to the superior light-harvesting efficiency of PANI nanofiber arrays. Water evaporation rates were measured at 3.13 kg m$^{-2}$ h$^{-1}$ under 3 sun illumination by recording the weight loss (Fig. 2i). This remarkable performance can be attributed to the high photothermal conversion efficiency and efficient water supply of the PANI nanoarrays evaporator.

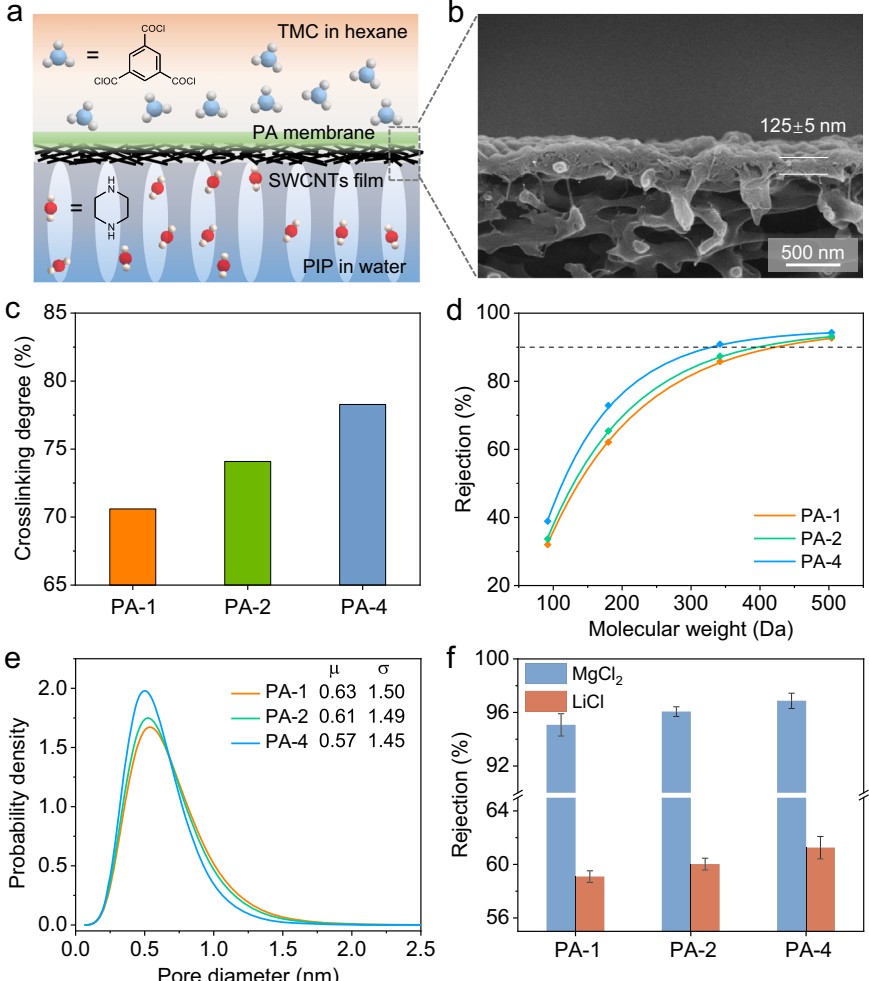

**Fig. 3 | Fabrication and characterization of PA membrane. a** Schematic illustration of the structure of SWCNTs film supported PA membrane. **b** Cross-sectional SEM images of the SWCNTs film supported PA membrane. **c** Crosslinking degree of PA membranes. **d** Rejections of neutral model solutes, including raffinose, sucrose, glucose, and glycerol by PA membranes. **e** Pore size probability distribution of the PA membranes, derived from the rejection curves of neutral solutes. **f** Rejections of $MgCl_2$ and LiCl by PA membranes with different crosslinking degrees. The experiments were conducted using a cross-flow filtration cell with an operating pressure of 4 bar and a cross-flow velocity of 3.0 cm/s. Error bar represents the standard deviation of three replicate measurements. Source data are provided as a Source Data file.

## Li$^+$/Mg$^{2+}$ separation performance of PA membrane

A membrane with high mono-/di-valent ion separation performance is highly required to extract lithium from brine effectively. State-of-the-art mono-/di-valent ion separation membranes are based on a thin film composite design, which deposits a PA selective layer, formed by interfacial polymerization between piperazine (PIP) and trimesoyl chloride (TMC), on the top of porous substrate. We have previously reported that deposition of a thin layer of single-walled carbon nanotubes (SWCNTs) film on macroporous PES membrane greatly improves the quality of the obtained PA layer thanks to the high porosity, smoothness, and narrowly distributed pores of the SWCNT film[53]. Following previous studies[24,53], PA membranes were fabricated on the surface of SWCNTs film via interfacial polymerization (Fig. 3a). Firstly, a thin SWCNT film was prepared through vacuum-filtrating a certain amount of SWCNTs dispersion onto a PES membrane with pore size of ~0.2 μm. The surface SEM image of SWCNTs film shows that a network porous structure was formed from interconnected carbon nanotubes with a high porosity (Supplementary Fig. 2a). Then a typical interfacial polymerization process was conducted on the surface of SWCNT film, in which PIP and TMC reacted at water-hexane interface to form the PA layer. A smooth PA layer was formed after the interfacial polymerization process using 2 mg mL$^{-1}$ TMC and 10 mg mL$^{-1}$ PIP solutions (Supplementary Fig. 2b).

The cross-sectional SEM image shows a bilayer structure with a thickness of 125 ± 5 nm (Fig. 3b).

The crosslinking degree of the resultant PA membrane reflects the efficiency of the interfacial reaction and determines the effective pore size. To study the effect of pore size on Li$^+$/Mg$^{2+}$ separation, the crosslinking degree of the PA layer was tuned by changing the TMC concentrations of 1, 2, to 4 mg mL$^{-1}$ during the interfacial polymerization process. The obtained membrane was denoted as PA-X, X representing the TMC concentration used in the interfacial polymerization process. The chemical composition of obtained PA layers was analyzed by X-ray photoelectron spectroscopy (XPS) (Supplementary Fig. 3). In the C1s XPS spectra, four peaks were detected, which were ascribed to O=C−O (centered at 288.5 eV), O=C−N (centered at 287.7 eV), C−N (centered at 286.0 eV), and C−C (centered at 284.8 eV) in the PA layer, further confirming the formation of the polyamide structure[24]. The O=C−N group was derived from the reaction between the acyl chloride in TMC monomers and secondary amine in PIP monomers. The O=C−O group can be attributed to the carboxyl acid group formed by the hydrolysis of the unreacted acylchloride group. With the TMC concentration increasing from 1 to 2 and 4 mg mL$^{-1}$, the O=C−N content gradually increased from 6.35% to 6.48%, and 7.26%. The O=C−O content decreased from 2.85% to 2.45% and 1.95% (Supplementary Table 1). The crosslinking degrees of PA layers were

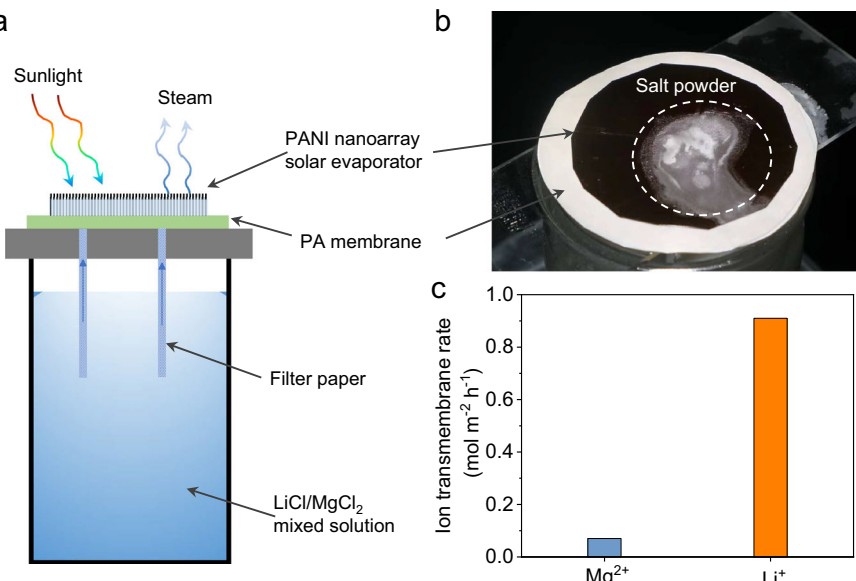

**Fig. 4 | Solar-driven lithium extraction device. a** Schematic illustration of the solar-driven Li$^+$/Mg$^{2+}$ separation device. The top is the PANI nanoarrays solar evaporator which evaporates water and generates capillary pressure to encourage a continuous upward flow of water and ions from bottom to surface. Beneath PANI nanoarrays solar evaporator, the PA-4 membrane is the selective barrier for Li$^+$/Mg$^{2+}$ separation. **b** Photograph of solid salt powders accumulated on the surface of the PANI nanoarrays solar evaporator. **c** Transmembrane rates of Li$^+$ and Mg$^{2+}$. Source data are provided as a Source Data file.

calculated based on the O/N elemental ratio from the XPS spectra. As shown in Fig. 3c, the crosslinking degrees of PA-1, PA-2, and PA-4 were 70.6%, 74.1%, and 78.3%, respectively. These results indicate that the increase in the TMC concentration could improve the crosslinking degree of the PA layer during the PIP-TMC interfacial polymerization process.

The molecular weight cut-off (MWCO) and pore size distribution of the PA membranes were calculated using neutral organic molecules as probes (detailed information see Supplementary Methods). As shown in Fig. 3d, the MWCO of the PA membrane decreased from 422 Da to 309 Da by increasing the TMC concentration from 1 to 4 mg mL$^{-1}$. According to the function fitting of the corresponding pore model, the mean pore size was 0.63, 0.61, and 0.57 nm for PA-1, PA-2, and PA-4, respectively (Fig. 3e). This result indicates that the effective pore size of the PA layer decreased with the increase of the crosslinking degree. The performance of the PA membrane was evaluated by experimenting with a cross-flow filtration with transmembrane pressure of 4.0 bar. The rejections to MgCl$_2$ (1.0 g L$^{-1}$) were 95.1%, 96.1%, and 96.9% for PA-1, PA-2, and PA-4, respectively. The rejection of LiCl (1.0 g L$^{-1}$) slightly increased from 59.1% to 61.3% with the increase of crosslinking degree. These results are reasonable when compared to the hydrated ion diameter and hydration energy of Li$^+$ and Mg$^{2+}$. The hydrated Mg$^{2+}$ diameter was 0.86 nm, larger than most of pores in the PA membrane. As Mg$^{2+}$ enters PA membrane, the energy required to strip water from the hydration shell of Mg$^{2+}$ is 437.4 kcal mol$^{-1}$[54]. In contrast, the hydrated Li$^+$ diameter is 0.76 nm, whereas the energy required to strip water from the hydration shell of Li$^+$ is 113.5 kcal mol$^{-1}$. Therefore, the PA membrane demonstrated a higher rejection rate to Mg$^{2+}$ than Li$^+$.

**Solar-driven lithium extraction**

After evaluating the basic properties of PANI nanoarrays solar evaporator and PA membrane, we proceeded the investigation of the separation performance of solar-driven Li$^+$/Mg$^{2+}$ separation. A solar-driven Li$^+$/Mg$^{2+}$ separation device was designed and constructed, which incorporated a PANI nanoarrays solar evaporator as the top photothermal layer and a PA-4 membrane as the selective barrier for Li$^+$/Mg$^{2+}$ separation (Fig. 4a). To evaluate Li$^+$/Mg$^{2+}$ separation

performance during interfacial evaporation, we employed a mixed solution containing 0.45 g L$^{-1}$ of LiCl and 0.90 g L$^{-1}$ of MgCl$_2$ (Li$^+$ and Mg$^{2+}$ had similar molality of ~100 mM). When the PA membrane was introduced beneath the PANI nanoarrays solar evaporator, the water evaporation rate slightly decreased to 2.95 kg m$^{-2}$ h$^{-1}$ under 3 sun irradiation. (Supplementary Fig. 4). After observing 120 min of stable evaporation, white salt powders accumulated on the surface of the PANI nanoarrays solar evaporator (Fig. 4b), indicating that capillary pressure enables water and ions permeation through the PA membrane. The transmembrane rate of Li$^+$ was found to be 0.91 mol m$^{-2}$ h$^{-1}$, which was 13 times faster than that of Mg$^{2+}$ in the ion separation membrane-based solar evaporator (Fig. 4c), indicating that the ultra-thin PA membrane maintains its monovalent/divalent ion selectivity, even under capillary pressure-driven separation process.

To investigate the effect of the Mg$^{2+}$/Li$^+$ ratio on solar-driven lithium extraction, the LiCl concentration was kept constant at 0.45 g L$^{-1}$ and the MgCl$_2$ concentration gradually increased from 0.45 to 4.50 g L$^{-1}$. Under 3 sun irradiation, the water evaporation rate remained around 2.9–3.0 kg m$^{-2}$ h$^{-1}$. However, it is observed that the LiCl crystallization rate increased from 37.3 to 61.8 g m$^{-2}$ h$^{-1}$ with the increase of Mg$^{2+}$/Li$^+$ ratio (Fig. 5a). This phenomenon can be explained by the Donnan equilibrium[55,56], where the increase in the Mg$^{2+}$/Li$^+$ ratio causes a proportional rise in Cl$^-$ concentration. With a higher amount of Cl$^-$ in the solution, more Cl$^-$ can permeate through the PA membrane. To maintain equilibrium on both sides of the membrane, Li$^+$ with a smaller hydrated ion diameter and lower hydration energy also trans through the PA membrane, increasing the LiCl crystallization rate. Figure 5b demonstrates the proportion of LiCl in solution and in salt powder collected on the PANI nanoarrays solar evaporator. During the evaporation process of MgCl$_2$/LiCl mixed solution with mass ratio of 1:1, the proportion of LiCl increased from 50.0% in the feed solution to 94.2% in the salt powder collected on the PANI evaporator surface. In the case of an MgCl$_2$/LiCl mixed solution with a mass ratio of 10:1, the proportion of LiCl was enriched from 9.1% in the feed solution to 52.4% in the salt powder, indicating a high lithium enrichment efficiency.

To further examine the effect of salt concentrations on solar-driven lithium extraction, the total concentration of MgCl$_2$ and LiCl

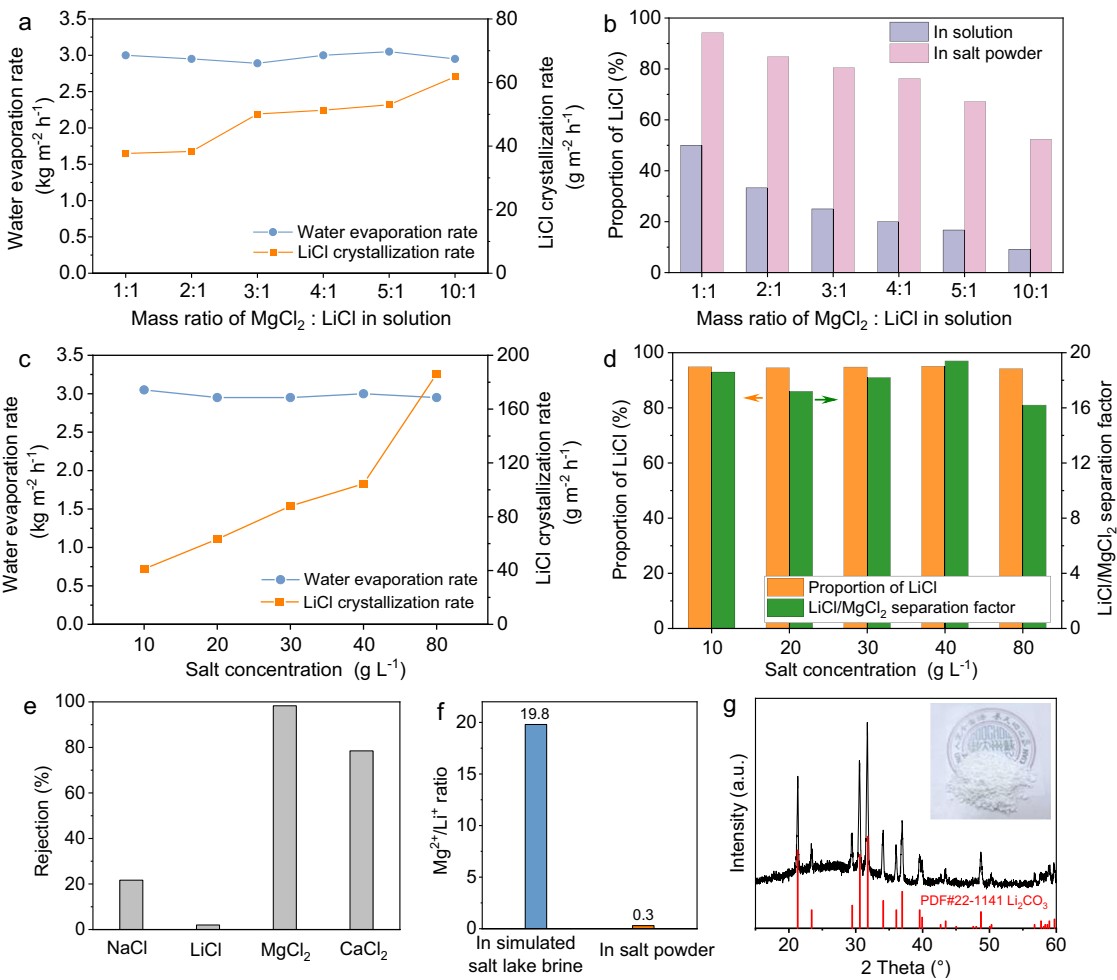

**Fig. 5 | Performance of solar-driven lithium extraction. a, b** the effect of the $Mg^{2+}/Li^+$ ratio on solar-driven lithium extraction. **c, d** the effect of salt concentrations on solar-driven lithium extraction. The water evaporation rate and salt crystallization rate were measured under 3 sun irradiation. **e** Salt rejection when evaporating a simulated salt-lake brine. **f** $Mg^{2+}/Li^+$ ratio in simulated salt-lake brine and solid salt powder collected on the solar evaporator, respectively. **g** XRD pattern of the collected sediment after adding $Na_2CO_3$. All diffraction peaks of the sediment matched with the standard $Li_2CO_3$ XRD pattern. Inset is a photograph of the collected sediment. Source data are provided as a Source Data file.

was increased from 10 to 80 g $L^{-1}$ while maintaining a $MgCl_2/LiCl$ mass ratio of 1:1. Under 3 sun irradiation, the water evaporation rate remained around 3.0 kg $m^{-2}$ $h^{-1}$, while the LiCl crystallization rate increased from 41.2 to 186.2 g $m^{-2}$ $h^{-1}$ (Fig. 5c). Notably, the PA membrane kept a stable $Mg^{2+}/Li^+$ separation performance, with the LiCl/$MgCl_2$ separation factor remaining above 16.2 (Fig. 5d). The proportion of LiCl in the salt powder collected from the surface of PANI nanoarrays solar evaporator was in the range of 94.2 to 95.1%, indicating that the device can effectively extract lithium ions from $Mg^{2+}/Li^+$ mixtures even in high concentration and high $Mg^{2+}/Li^+$ ratio.

The device was further exploited to process a simulated salt-lake brine containing 268.0 g $L^{-1}$ NaCl, 5.1 g $L^{-1}$ LiCl, 66.1 g $L^{-1}$ $MgCl_2$, and 9.2 g $L^{-1}$ $CaCl_2$ (referring to the compositions of the Uyuni salar brine[57,58]). In the simulated salt-lake brine, total salt concentration was as high as 348.4 g $L^{-1}$ and $Mg^{2+}/Li^+$ mass ratio was up to 19.8. Under 3 sun irradiation, the water evaporation rate was around 2.2 kg $m^{-2}$ $h^{-1}$, and salt crystallization rate was 480 g $m^{-2}$ $h^{-1}$. The salt powder can be directly collected from the surface of the PANI nanoarrays solar evaporator. The composition was analyzed by inductively coupled plasma atomic emission spectroscopy. Compared with the composition of simulated salt-lake brine, NaCl proportion in the solid salt powder increased from 76.9% to 96.3%, LiCl proportion increased from 1.5% to 2.3% (Supplementary Fig. 5). Correspondingly, $MgCl_2$ proportion

decreased from 19% to 0.5% and $CaCl_2$ proportion decreased from 2.7% to 0.9%. These results indicate that monovalent salts can be enriched by the synergistic effect of membrane-based ion separation and solar-driven evaporation. To determine the salt rejection, we assume that the salt powder was dissolved in the evaporated water. The rejection to $MgCl_2$ was as high as 98.3%, and the rejection to $CaCl_2$ was 78.5% (Fig. 5e and Supplementary Table 2). The slightly lower rejection to $CaCl_2$ than that of $MgCl_2$ is ascribed to the smaller hydrated diameter of $Ca^{2+}$ than $Mg^{2+}$. In contrast, the rejection to NaCl and LiCl was 21.7% and 2.0%, respectively, indicating that monovalent salt ions, like $Na^+$ and $K^+$, can pass through the PA membrane, while most of the $Mg^{2+}$ and $Ca^{2+}$ ions were rejected.

The ratio of magnesium and lithium in brines is a key factor in determining the process of lithium extraction. When the $Mg^{2+}/Li^+$ mass ratio is lower, it becomes easier to extract lithium from salt-lake brines. The mass ratio of $Mg^{2+}/Li^+$ in the salt powders collected on the surface of the PANI nanoarrays solar evaporator was 0.3. This is a significant decrease compared to the simulated salt-lake brine, with a reduction of 66 times (Fig. 5f). At a low $Mg^{2+}/Li^+$ ratio, after adding NaOH and sodium oxalate to remove residual small amounts of $Mg^{2+}$ and $Ca^{2+}$, lithium can be readily precipitated out in the form of $Li_2CO_3$ by adding $Na_2CO_3$. The sediment was separated by centrifugation, washed using deionized water, and then dried in a vacuum oven. The collected white

powder (Fig. 5g inset) was characterized by powder X-ray diffraction spectroscopy (XRD), whereby the pattern fitted well with the standard pattern of $Li_2CO_3$ (PDF#22-1141) without any impurity signals being detected. Further quantitative elemental analysis shows that the purity of $Li_2CO_3$ was around 99% which meets the requirements of battery-grade $Li_2CO_3$ purity.

To evaluate the potential for application, we carried out an outdoor experiment which carried out from 9:00 to 16:00 under natural sunlight with an average solar heat flux of ~0.7 kw m$^{-2}$. The membrane-based solar evaporator was employed to treat the simulated salt-lake brine. As shown in Supplementary Fig. 6a, the rate of water evaporation was influenced by the intensity of solar irradiation. The maximum water evaporation rate of 0.95 L m$^{-2}$ h$^{-1}$ was observed at noon, which was consistent with the result obtained in the laboratory under 1 sun irradiation. At 16:00, the salt crystals were collected, and the PANI was regenerated for tomorrow's experiment. During the following 5-day test (Supplementary Fig. 6b), the amount of evaporated water and collected salt crystals were maintained stable, showing its great potential for practical lithium extraction. In addition, the scalability of the membrane-based solar evaporator is also a critically important consideration. The system can be scaled up easily by increasing its size and multiplying the brine treatment capacity using an array of solar evaporators.

In summary, drawing inspiration from the selective water/ion uptake and salt secretion processes in mangroves, we reported the synergistic design of ion separation membrane and solar-driven interfacial evaporator, which effectively extracts solid LiCl from a mixed salt solution, achieving a purity level of up to 94%. Notably, our ion separation membrane-based solar evaporator demonstrates the ability to directly treat simulated salt-lake brine, even at concentrations as high as 348.4 g L$^{-1}$, resulting in a remarkable 66-fold reduction in the $Mg^{2+}/Li^+$ ratio (from 19.8 to 0.3). At a low $Mg^{2+}/Li^+$ ratio, battery-grade $Li_2CO_3$ powder can be precipitated out through a simple precipitation process. By combining ion separation with solar-driven interfacial evaporation, our research introduces a new and promising approach for lithium extraction utilizing renewable energy sources. Hence, it is expected that this approach will lead to the development of a promising process to secure the lithium supply for future energy uses.

## Methods

### Fabrication of PANI nanoarrays evaporator
Initially, 93 mg aniline and 114 mg ammonium persulfate were dissolved separately in 50 mL of 1.0 mol L$^{-1}$ perchloric acid solution under magnetic stirring. These were then cooled in an ice-water bath for 30 min, resulting in a 0.02 mol L$^{-1}$ aniline solution and a 0.01 mol L$^{-1}$ ammonium persulfate solution. Subsequently, the ammonium persulfate solution was rapidly poured into the aniline solution, and a PES membrane was submerged in the mixture. The reaction proceeded with magnetic stirring in an ice-water bath for 12 h. Upon completion of the reaction, the membrane was rinsed with deionized water and soaked in a 0.1 mol L$^{-1}$ ammonium hydroxide solution for 2 h. Afterward, the membrane was thoroughly rinsed with deionized water and dried in a forced-air oven at 60 °C.

### Fabrication of PA membrane
The polyamide membrane was prepared at 25.0 ± 0.5 °C and 60 ± 5% relative humidity. 1.0 g PIP and 0.24 g sodium dodecyl sulfate were dissolved in 100 mL water to produce PIP aqueous solution. The SWCNTs film was first placed on a glass plate and then impregnated with PIP solution for 30 s. The glass plate was drained vertically, and excess PIP solution was removed from the SWCNTs film surface. Then TMC n-hexane solution (1, 2, and 4 mg mL$^{-1}$) was poured onto the surface of SWCNTs film for 30 s which resulted in the formation of a polyamide active layer. The resulting SWCNTs film supported PA

membrane was immersed in n-hexane for 30 s to remove unreacted TMC, then heated in an oven at 90 °C for 30 min to increase the crosslinking degree of polyamide network. The membrane after heating was placed in deionized water and stored in a refrigerator at 4 °C.

### Material characterizations
The morphology of the PANI nanoarrays evaporator and PA membranes were captured by a field emission SEM (Hitachi S8230). The light absorption property of PANI-m was analyzed by ultraviolet and visible spectrophotometry (UV2600, Shimadzu). The infrared images and surface temperatures were captured by an ICI 8320 infrared camera (ICI, USA). The water CAs were measured on an OCA20 machine (Data-physics, Germany). The average pore size of PES substrate and PANI nanoarrays evaporator were measured by a bubble-pressure method membrane pore size analyzer (Beishide, 3H-2000PB). The surface elements of the polyamide membranes were analyzed by X-ray photoelectron spectroscopy (Thermo Scientific EXCALAB 250).

### Evaluation of Li$^+$/Mg$^{2+}$ separation of PA membrane
The separation performance of PA membrane was evaluated by using a cross-flow filtration apparatus at room temperature. The effective area of PA membrane was 7.1 cm$^2$. Either 1.0 g L$^{-1}$ LiCl solution or 1.0 g L$^{-1}$ $MgCl_2$ solution was used as feed solution. Rejection was calculated based on the Eq. (1). The concentration of each solution was obtained by conductivity.

$$R = \left(1 - \frac{c_p}{c_f}\right) \times 100\% \tag{1}$$

Where $c_f$ and $c_p$ represent the concentration of feed and filtrate solutions, respectively.

### Evaluation of solar-driven lithium extraction
The solar-driven lithium extraction was conducted at a temperature of 25.0 ± 0.5 °C and relative humidity of 40 ± 5%. A solar simulator (Xe lamp, PL-X300, Beijing Precise Technology Co., Ltd.) was used for the indoor tests. The test device for solar-driven lithium extraction is shown in Fig. 4a. A LiCl and $MgCl_2$ mixed solution was added in a container. When evaluating the effect of the $Mg^{2+}/Li^+$ ratio on solar-driven lithium extraction, the LiCl concentration was kept constant at 0.45 g L$^{-1}$ in the mixed salt solution while the $MgCl_2$ concentration was gradually increased from 0.45 to 4.50 g L$^{-1}$. When investigating the impact of salt concentrations, the overall salt concentration was increased from 10 to 80 g/L while keeping a $MgCl_2$/LiCl mass ratio of 1.0. The solution was conducted through a hydrophilic filter paper to the PA membrane. To ensure that the ions pass through the PA membrane for selective separation and then reach the PANI nanoarrays evaporator, the diameter of the PA membrane is larger than the PANI nanoarrays evaporator, with diameters of 4.0 cm and 3.0 cm, respectively. The mass change of the mixed solution and PANI nanoarrays evaporator were recorded by a high-accuracy electronic balance (METTLER TOLEDO ML204, 0.1 mg accuracy) and used to calculate the water evaporator rate and salt crystallization rate, respectively. The salt powder was collected and redissolved in water to measure the LiCl and $MgCl_2$ content by inductively coupled plasma atomic emission spectroscopy (ICP-OES, Agilent 5110).

The LiCl/$MgCl_2$ separation factor was calculated by the following Eq. (2):

$$S_{LiCl/MgCl_2} = \frac{\left(C_{LiCl}/C_{MgCl_2}\right)_{in\,powder}}{\left(C_{LiCl}/C_{MgCl_2}\right)_{in\,solution}} \tag{2}$$

Where $C_{LiCl}$ and $C_{MgCl_2}$ represent the concentration of LiCl and $MgCl_2$, respectively.

## Lithium extraction from simulated salt-lake brine

The bine was prepared based on the typical geothermal brine composition at Uyuni salar brine[57,58], in Bolivia, containing 268.0 g L$^{-1}$ NaCl, 5.1 g L$^{-1}$ LiCl, 66.1 g L$^{-1}$ MgCl$_2$, and 9.2 g L$^{-1}$ CaCl$_2$. The simulated salt-lake brine was used as the feed solution for the solar evaporator. Under continuous irradiation, salt crystals formed on the PANI layer surface and were collected once the surface was fully covered. 5 g of collected salt powders were added in 5 mL water to create an over-saturated solution. The pH value was adjusted to 11 using a saturated NaOH solution to remove any remaining Mg$^{2+}$. The white precipitate was removed by centrifuge at 1467 g for 10 mins. Next, a saturated sodium oxalate solution was slowly added to the upper clear solution in order to remove any remaining small amount of Ca$^{2+}$. The mixture was centrifuged again at 1467 g for 10 mins, and the upper clear solution was collected. Finally, a saturated Na$_2$CO$_3$ solution was added to the collected solution at 80 °C. The white precipitates were collected after centrifuge and dried in an air-forced oven at 60 °C for 12 h.

## Data availability

All data supporting the findings of this study are available in the article, the Supplementary Information and the Source Data file. Source data are provided with this paper.

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

## Acknowledgements

The authors gratefully acknowledge the support from the National Key Research and Development Program of China (2022YFB3805900, 2022YFB3805903, 2023YFE0113700), the National Natural Science Foundation of China (21988102, 22208229, 22072104), the Key Research and Development Plan of Jiangsu Province (BE2022056), the Natural Science Foundation of Jiangsu Province (BK20220501), Gusu Innovation and Entrepreneurship Leading Talent Plan (ZXL2023198).

## Author contributions

S.Z., J.L. and J.J. designed the experiments. X.W., S.Z. and X.C. performed the experiments including the fabrication of the PANI evaporator and PA membranes, characterization, and performance test. M.W. and M.P. provided suggestions and technical support. S.Z. and J.J. organized the data and wrote the paper. All authors discussed the results and approved the final version of the article.

## Competing interests

The authors declare no competing interests.
