## [Peer Review File · Nature Communications]

Solar-driven membrane separation for direct lithium extraction from artificial salt-lake brineReviewers' Comments:

Reviewer #1 (Remarks to the Author):

Authors prepared three-layer membranes (so called PANI nanoarrays solar evaporator) for Li ion separation from high concentrated artificial brine solution. Nanofiltration membranes have been known to reject divalent ions and pass monovalent ions so that they can separate Mg and concentrate Li ions. One of the disadvantages of NF is to dilute the concentrated brine and after separation, they have to concentrated again. In the present manuscript, authors proposed the solar-driven evaporation and NF technique to eliminate the dilemma of NF by adding PANI nanoarrays. This design is quite unique, and authors present that this process lithium extraction capacity of 94%. This way, lithium extraction can be efficiently conducted. However, before acceptance of this manuscript, the following concerns need to be addressed.

- (1) Title: "Synergistic design of membrane-based ion ...from salt-lake brine". It impresses at the beginning that authors may have used the actual salt-lake brine, but they used the artificial brine. Reviewer suggests adding "artificial" before salt-lake.
- (2) Authors may need to address that by using three-layer membranes for Li extraction, how fast it is to extract Li, compared with the solar evaporation itself, or NF or other methods.
- (3) Regarding the PANI layer on PES membrane, how fast is the reaction?
- (4) Figure 2: There is no description on Figure 2.

Overall, the manuscript is well prepared and described to publish in Science Advances.

Reviewer #2 (Remarks to the Author):

This study has reported the direct extraction of LiCl powder from simulated salt-lake brine by combining an ion separation membrane and a solar-driven evaporator system, inspired by the selective water/ion uptake and salt secretion in mangroves. The process involves an upper polyaniline (PANI) nanofiber array as a photothermal layer to evaporate water, a middle hydrophilic PES macroporous membrane generating capillary pressure as the driving force for water transport, and a bottom ultrathin polyamide ion separation membrane allowing passage of lithium ions while rejecting other multivalent ions, like calcium and magnesium. The system can collect LiCl powder with a purity of 94.2% from a LiCl/MgCl₂ mixed solution. In treating with the simulated salt-lake brine with a TDS of 348.4 g L⁻¹, the process has the ability to reduce the Mg²⁺/Li⁺ ratio from 19.8 to 0.3. The idea of lithium extraction by bioinspiring from mangroves is interesting. However, the paper doesn't meet the standards that make it improper to publish in Nature Communications.

1. Although the concept seems new, the commercial potential of this study is under question. It seems the system has not demonstrated monovalent selectivity Na/Li. The Mg can be easily separated from Li, on a large scale, using more conventional unit operations that have much less complexity than this work,

even an NF system would be helpful. The study claims that it has the ability to produce Li_2CO_3 with battery-grade purity. However, according to the crystallization rate of LiCl , nearly all LiCl and NaCl can pass through the system and form a crystal mixture of NaCl and LiCl . Based on the concentration of LiCl and NaCl in the simulated lake brine, the amount of NaCl salt is much higher than LiCl . How is it claimed that the study can produce Li_2CO_3 with a purity of 99%? It is noteworthy to mention the study has not conducted any experiment regarding the precipitation process to purify LiCl and extract it from NaCl salts.

2. Crystal formation can occur within the photothermal layers, leading to the clogging of membrane pores and blocking the photothermal layer. The study lacks a fouling/scaling control strategy and does not employ scaling prediction methods.

3. The diffusion rate of lithium ions hasn't been compared with the LiCl crystallization rate. From Fig. 4c, the transmembrane rate of lithium ($0.9 \text{ mol m}^{-2} \text{ h}^{-1}$) is calculated as around $6.2 \text{ g m}^{-2} \text{ h}^{-1}$, which is much lower than the LiCl crystallization rate (approximately $40 \text{ g m}^{-2} \text{ h}^{-1}$, Fig. 5a). The diffusion rate and the crystallization rate don't align with each other in this study. A mass balance needs to be considered.

4. The selectivity of Li over other ions has not been evaluated.

Please find below the general comments:

1. The abstract exceeds the recommended words outlined in the guide for authors (approximately 220 words).

2. Line 95, Fig. 1 a doesn't match with the context. It should be Figure 2. (Same for Fig. 1 b-i from line 103 to line 135)

3. Use consistent terminology for nanofiber. In line 99, change nano-fibers to nanofibers.

4. It hasn't been noted why PANI and PES are chosen.

5. The introduction requires revision in some parts as it doesn't adequately introduce the main question that the study aims to address. Also, it lacks a literature review and does not showcase the novelty of this research, which is crucial to highlight the significance of the study.

6. It's not common to indicate the results in the introduction, i.e., line 88: with a purity as high as 94.5%.

7. What is the reference for the method of Fabrication of PANI nanoarrays evaporator? For example, why are 93 mg aniline and 114 mg ammonium persulfate chosen to fabricate the nanoarrays?

8. Refer to the previous comment for the fabrication of the PA membrane.

9. In line 328, what do you mean by the concentration of LiCl , MgCl_2 feed solution were 2000 mg L^{-1} , respectively? Is the concentration of both LiCl and MgCl_2 equivalent to 2000 mg L^{-1} ? In the results, their concentration has been reported as 1 mg L^{-1} for rejection tests.

10. It's not apparent in the method that the membrane performance has been tested in the mixed solution. All the procedures (which are explained in the results in the current format) must be moved to their related sections in the method.

11. It had to be mentioned how the required energy for stripping water from the hydration shell of Mg and Li is calculated. (Lines 195-197)

12. Use relevant references on Donnan equilibrium to support the discussion made on the increase in LiCl crystallization rate from 37.3 to $61.8 \text{ g m}^{-2} \text{ h}^{-1}$.

13. There is no discussion in the context of paper for Fig. 5a.

14. How did you find the concentrations in the Uyuni salar brine? Why didn't you add KCl to the simulated brine? It doesn't have any reference.

Reviewer #1 (Remarks to the Author):

General comment: Authors prepared three-layer membranes (so called PANI nanoarrays solar evaporator) for Li ion separation from high concentrated artificial brine solution. Nanofiltration membranes have been known to reject divalent ions and pass monovalent ions so that they can separate Mg and concentrate Li ions. One of the disadvantages of NF is to dilute the concentrated brine and after separation, they have to concentrated again. In the present manuscript, authors proposed the solar-driven evaporation and NF technique to eliminate the dilemma of NF by adding PANI nanoarrays. This design is quite unique, and authors present that this process lithium extraction capacity of 94%. This way, lithium extraction can be efficiently conducted. However, before acceptance of this manuscript, the following concerns need to be addressed.

Response: We gratefully appreciate the reviewer's careful reading and positive comments to our manuscript. The manuscript has been thoroughly revised according to the reviewer's insightful suggestions as detailed below. All changes have been highlighted in red in the revised manuscript.

(1) Title: "Synergistic design of membrane-based ion ...from salt-lake brine". It impresses at the beginning that authors may have used the actual salt-lake brine, but they used the artificial brine. Reviewer suggests adding "artificial" before salt-lake.

Response: Thanks for the reviewer's valuable comment. We would like to point out that the simulated salt-lake water was prepared by referring to the composition of actual salt-lake brine. For example, in Figure 5e-g, the Mg^{2+}/Li^+ ratio and total salt concentration were determined according to the composition of the Uyuni salar brine (*J. Membr. Sci.* **2021**, 635, 119441; *Hydrometallurgy* **2012**,117–118, 64). To clarify this point, we have added the word "artificial" before "salt-lake brine" in the revised title as suggested by the reviewer.

(2) Authors may need to address that by using three-layer membranes for Li extraction, how fast it is to extract Li, compared with the solar evaporation itself, or NF or other methods.

Response: Solar evaporation is commonly used as the first step in lithium extraction. Brine is concentrated in ponds, leading to precipitation of NaCl and KCl salts. After the precipitation of KCl, Li^+ concentration in raw brine is enriched to 0.3-0.5 g L⁻¹ or even

2-7 g L⁻¹ depending on the process design and original brine concentration (*Miner. Eng.* **2016**, *89*, 119; *Adv. Sci.* **2022**, *9*, 2201380). However, Mg²⁺ is enriched at the same time, close to saturation in some cases. Thus, solar evaporation itself cannot achieve efficient Li⁺/Mg²⁺ separation especially for high Mg/Li ratio brines.

Nanofiltration (NF) is the most widely adopted Li⁺/Mg²⁺ separation approach in the salt-lake lithium extraction industry. Because of the high osmotic pressure of highly concentrated brines, an NF membrane requires a high applied pressure that may exceed the mechanical strength of the membrane module. Therefore, the conventional NF process typically involves the 10-20 times dilution of high-concentration brine with a large quantity of freshwater in the pre-treatment stage. Moreover, NF membrane separation processes solely yield Li⁺-enriched solutions, necessitating further concentration to obtain solid lithium products, which requires energy-intensive processes like thermal distillation or high-pressure RO to obtain the Li⁺-enriched solution with the required concentration. Despite its higher efficiency, the NF process requires at least 4000 kWh of electricity to generate one ton of Li₂CO₃ (*J. Membr. Sci.* **2021**, *635*, 119441).

In this work, we designed a new method utilizing the synergistic effect of an ion separation membrane and a solar-driven evaporator to realize Li⁺/Mg²⁺ separation and Li⁺ concentration in one step. The ion separation membrane under the photothermal layer allows Li⁺ to pass through and block Mg²⁺. During the water evaporation process, Li⁺ was selectively enriched in the photothermal layer and crystallized in the form of LiCl. Our designed ion separation membrane-based solar evaporator has a water evaporation rate of around 1.0 kg m⁻² h⁻¹ under 1 sun irradiation. We would like to clarify that the efficiency of solar-powered evaporation can not compete with energy-intensive industrial processes. The advantage of our designed membrane-based solar evaporator is that it can realize Li⁺/Mg²⁺ separation and Li⁺ concentration in one step only utilizing the abundant solar energy resources in the salt-lake area, which offers a low-barrier-of-entry option for lithium extraction.

(3) Regarding the PANI layer on PES membrane, how fast is the reaction?

Response: According to the reviewer's comment, we monitored the polymerization process of aniline in solution. There was no color change for the first 3 hours, then the solution gradually turned light blue and darkened to a black-blue color over the next 60 minutes (Figure R1). These results suggest that aniline polymerization consists of two

stages: nucleation and growth (*Angew. Chem. Int. Ed.* **2004**, *43*, 5817; *Adv. Mater.* **2005**, *17*, 1679). The first three hours involve nucleation, followed by a rapid growth stage once nuclear sites have formed. To clarify this point, the sentence “The polymerization process formed nano-fibers by rapidly polymerizing aniline monomers via ammonium persulfate.” in the original manuscript has been revised to “Nanofibers were formed through a polymerization process of aniline monomers using ammonium persulfate.” (page 5, line 20-22).

Figure R1. Photographs of the polymerization process of aniline in solution.

(4) Figure 2: There is no description on Figure 2.

Response: We are sorry for this mistake. The part “Fabrication and characterization of PANI nanoarrays solar evaporator (Page 5-7)” is the description of Figure 2. We have revised them in the revised manuscript.

Reviewer #2 (Remarks to the Author):

General comment: This study has reported the direct extraction of LiCl powder from simulated salt-lake brine by combining an ion separation membrane and a solar-driven evaporator system, inspired by the selective water/ion uptake and salt secretion in mangroves. The process involves an upper polyaniline (PANI) nanofiber array as a photothermal layer to evaporate water, a middle hydrophilic PES macroporous membrane generating capillary pressure as the driving force for water transport, and a bottom ultrathin polyamide ion separation membrane allowing passage of lithium ions while rejecting other multivalent ions, like calcium and magnesium. The system can

collect LiCl powder with a purity of 94.2% from a LiCl/MgCl₂ mixed solution. In treating with the simulated salt-lake brine with a TDS of 348.4 g L⁻¹, the process has the ability to reduce the Mg²⁺/Li⁺ ratio from 19.8 to 0.3. The idea of lithium extraction by bioinspiring from mangroves is interesting. However, the paper doesn't meet the standards that make it improper to publish in *Nature Communications*.

Response: We are very grateful to the reviewer's comments and suggestions, which are beneficial to improve the quality of our manuscript. Meanwhile, we appreciate the reviewer for pointing out some problems, which are all very important for revising our manuscript. We have tried our best to conduct additional experiments and provide further discussions to address the comments of the reviewer. All changes have been highlighted in red color in the revised manuscript. We hope that the revised manuscript is suitable for publication in *Nature Communications*.

(1) Although the concept seems new, the commercial potential of this study is under question. It seems the system has not demonstrated monovalent selectivity Na/Li. The Mg can be easily separated from Li, on a large scale, using more conventional unit operations that have much less complexity than this work, even an NF system would be helpful. The study claims that it has the ability to produce Li₂CO₃ with battery-grade purity. However, according to the crystallization rate of LiCl, nearly all LiCl and NaCl can pass through the system and form a crystal mixture of NaCl and LiCl. Based on the concentration of LiCl and NaCl in the simulated lake brine, the amount of NaCl salt is much higher than LiCl. How is it claimed that the study can produce Li₂CO₃ with a purity of 99%? It is noteworthy to mention the study has not conducted any experiment regarding the precipitation process to purify LiCl and extract it from NaCl salts.

Response: We would like to clarify that lithium extraction from salt-lake brine mainly involves three stages. In the first stage, brine will be concentrated in solar evaporation ponds, resulting in the precipitation of NaCl and KCl salts. After the precipitation of NaCl and KCl salts, Li⁺ concentration in raw brine is enriched to 0.3-0.5 g L⁻¹ or even 2.0-7.0 g L⁻¹ depending on the process design and original brine concentration (*J. Membr. Sci.* **2021**, 635, 119441; *Miner. Eng.* **2016**, 89, 119; *Adv. Sci.* **2022**, 9, 2201380). In some cases, the concentrations of Na⁺ and K⁺ ions are similar to that of lithium ions in raw brine. Meanwhile, Mg²⁺ ions are enriched and can reach saturation in certain cases. Therefore, reducing the Mg²⁺ at the second stage is critical for lithium carbonate production. There is no doubt that NF membrane separation technology is one of the

effective ion separation methods, particularly for $\text{Li}^+/\text{Mg}^{2+}$ separation. Although, all of the monovalent ions can pass through the NF membrane, the separation of monovalent ions will be realized in the last stage of lithium extraction from brine by adding Na_2CO_3 to form Li_2CO_3 precipitation. The solubility of Li_2CO_3 , Na_2CO_3 , and K_2CO_3 are quite different, where the solubility of Li_2CO_3 is very low (0.82 at 80 °C) and the solubility of Na_2CO_3 and K_2CO_3 are high (45.1 at 80 °C and 140 at 80 °C, respectively). Therefore, Li^+ can be easily precipitated in the form of carbonates after adding Na_2CO_3 .

In our original manuscript, we have mentioned that “At a low $\text{Mg}^{2+}/\text{Li}^+$ ratio, after adding NaOH and sodium oxalate to remove residual small amounts of Mg^{2+} and Ca^{2+} , lithium can be readily precipitated out in the form of Li_2CO_3 by adding Na_2CO_3 .” To clarify this point, the detailed method of carbonate precipitation was added in Method section in the revised manuscript (page 16, line 21-25 and page 17, line 1-9). We copied here for reviewer’s convenience:

“5 g of collected salt powders were added in 5 mL water to create a saturated solution. The pH value was adjusted to 11 using a saturated NaOH solution to remove any remaining Mg^{2+} . The white precipitate was removed by centrifuge at 5000 rpm for 10 mins. Next, a saturated sodium oxalate solution was slowly added to the upper clear solution in order to remove any remaining small amount of Ca^{2+} . The mixture was centrifuged again at 5000 rpm for 10 mins, and the upper clear solution was collected. Finally, a saturated Na_2CO_3 solution was added to the collected solution at 80 °C. The white precipitates were collected after centrifuge and dried in an air-forced oven at 60 °C for 12 hours.”

(2) Crystal formation can occur within the photothermal layers, leading to the clogging of membrane pores and blocking the photothermal layer. The study lacks a fouling/scaling control strategy and does not employ scaling prediction methods.

Response: We understand the reviewer’s concern. A 6-hour continuous experiment was conducted to evaporate simulated salt-lake brine with a TDS of 348.4 g L^{-1} under 1 sun irradiation. The experiment recorded the water evaporation rate, surface temperature of PANI photothermal layer, and salt crystal formation process (Figure R2). At the beginning, the water evaporation rate is $1.08 \text{ L m}^{-2} \text{ h}^{-1}$, the surface temperature of PANI photothermal layer is 41 °C. At the 3rd hour, tiny salt crystals can be observed on the surface of PANI photothermal layer. The water evaporation rate slightly declines to $0.95 \text{ L m}^{-2} \text{ h}^{-1}$. With the increasing evaporation time, more and more salt crystals can

be observed on the surface. It is worth noting that the surface temperature was maintained at 41-42 °C even though the surface was fully covered by salt crystals. After 6 hours, the rate of water evaporation decreases to 0.90 L m⁻² h⁻¹, which suggests that the salt crystals formed on the surface of the photothermal layer are affecting the water evaporation. However, in this structure, the salt crystals are unable to completely block the membrane pores or the photothermal layer.

The target of our membrane-based solar evaporator is obtaining Li-enriched salt powder. We propose a fouling control strategy to regenerate the PANI photothermal layer by recovering salt crystals during weak sun irradiation or at night. The photothermal layer and the ion separation layer are separate from each other (Figure R2b). This means that when the photothermal layer is fully covered by salt powder, it can be removed from the evaporator. Large salt crystals can be removed using a scraper, while any remaining small salt crystals can be washed away with a small amount of water. We will utilize the collected crystals and solution for the precipitation of lithium carbonate. After regenerating the PANI photothermal layer, the water evaporation rate can recover to the initial value of around 1.0 L m⁻² h⁻¹.

Figure R2. (a) Water evaporation rate and surface temperature of PANI photothermal layer during the 6-hour continuous experiment for evaporating simulated salt-lake brine with a TDS of 348.4 g L^{-1} under 1 sun irradiation. (b) Photography of salt crystal formation process on the surface of PANI photothermal layer. After the 6-hour continuous experiment, large salt crystals can be removed using a scraper, while any remaining small salt crystals can be washed away with a small amount of water to regenerate the PANI photothermal layer.

To evaluate the potential for application, we carried out an outdoor experiment from 9:00 to 16:00 under natural sunlight with an average intensity of $\sim 700 \text{ W m}^{-2}$. The membrane-based solar evaporator was employed to treat the simulated salt-lake brine. As shown in Figure R3a, the rate of water evaporation is influenced by the intensity of solar irradiation. The maximum water evaporation rate of $0.95 \text{ L m}^{-2} \text{ h}^{-1}$ was observed at noon, which is consistent with the result obtained in the laboratory under 1 sun irradiation. At 16:00, the salt crystals were collected, and the PANI was regenerated for tomorrow's experiment. During the following 5-day test, the amount of evaporated water and collected salt crystals were maintained stable (Figure R3b), showing its great potential for practical lithium extraction.

Figure R3. (a) Sunlight intensity and water evaporation rate during the outdoor experiment from 9:00 to 16:00 at Soochow University campus during 25-29th October. Inset in Figure R4a is the setup for the evaporation. (b) The amount of evaporated water and collected salt crystals on the unit area during the 5-day test.

In addition, the scalability of the membrane-based solar evaporator is also a critically important consideration. The system can be scaled up easily by increasing its size and multiplying the brine treatment capacity using an array of solar evaporators. Under the

field conditions, the solar incident angle, temperature, relative humidity, and natural wind will affect the performance of the solar evaporator. A comprehensive evaluation of such effects is necessary for the practical application of the solar crystallizer toward practical lithium extraction.

Figure R3 has been added to the revised Supplementary Materials as Supplementary Figure 6, and the corresponding discussion has been added in the revised manuscript (page 13, line 2-15). We copied here for reviewer's convenience:

“To evaluate the potential for application, we carried out an outdoor experiment which carried out from 9:00 to 16:00 under natural sunlight with an average solar heat flux of $\sim 0.7 \text{ kW m}^{-2}$. The membrane-based solar evaporator was employed to treat the simulated salt-lake brine. As shown in Supplementary Fig. 6, the rate of water evaporation is influenced by the intensity of solar irradiation. The maximum water evaporation rate of $0.95 \text{ L m}^{-2} \text{ h}^{-1}$ was observed at noon, which is consistent with the result obtained in the laboratory under 1 sun irradiation. At 16:00, the salt crystals were collected, and the PANI was regenerated for tomorrow's experiment. During the following 5-day test, the amount of evaporated water and collected salt crystals were maintained stable, showing its great potential for practical lithium extraction. In addition, the scalability of the membrane-based solar evaporator is also a critically important consideration. The system can be scaled up easily by increasing its size and multiplying the brine treatment capacity using an array of solar evaporators.”

(3) The diffusion rate of lithium ions hasn't been compared with the LiCl crystallization rate. From Fig. 4c, the transmembrane rate of lithium ($0.9 \text{ mol m}^{-2} \text{ h}^{-1}$) is calculated as around $6.2 \text{ g m}^{-2} \text{ h}^{-1}$, which is much lower than the LiCl crystallization rate (approximately $40 \text{ g m}^{-2} \text{ h}^{-1}$, Fig. 5a). The diffusion rate and the crystallization rate don't align with each other in this study. A mass balance needs to be considered.

Response: In Fig. 4c, we measured the transmembrane rate of lithium at $0.9 \text{ mol m}^{-2} \text{ h}^{-1}$ when evaporating a mixed solution containing 0.45 g L^{-1} of LiCl. The relative molecular mass of LiCl is 42.39 g mol^{-1} . The LiCl crystallization is calculated to be $38.2 \text{ g m}^{-2} \text{ h}^{-1}$ (0.9×42.39). Therefore, the diffusion rate is consistent with the crystallization.

(4) The selectivity of Li over other ions has not been evaluated.

Response: The selectivity of Li^+ over K^+ , Na^+ , Ca^{2+} and Mg^{2+} were evaluated by comparing their transmembrane rate. We measured the transmembrane ion rate for chloride salt solutions at 10 mmol L^{-1} under 3 sun irradiation. The rates for Li^+ , K^+ , Na^+ , Ca^{2+} , and Mg^{2+} were 0.91, 0.86, 0.84, 0.15, and $0.07 \text{ mol m}^{-2} \text{ h}^{-1}$, respectively (Figure R4a). The selectivity of Li^+/K^+ , Li^+/Na^+ , $\text{Li}^+/\text{Ca}^{2+}$, $\text{Li}^+/\text{Mg}^{2+}$ are 1.06, 1.08, 6.07 and 13.00, respectively (Figure R4b). Here, we would like to clarify that although the selectivity of Li^+/K^+ and Li^+/Na^+ are close to 1, it is not an issue to be solved by NF membrane process or membrane-based solar evaporation process, because the separation of monovalent ions will be realized in the last stage of lithium extraction from brine by adding Na_2CO_3 to form Li_2CO_3 precipitation as we have explained in response to reviewer2's comment #1. The solubility of Li_2CO_3 , Na_2CO_3 , and K_2CO_3 are quite different, where the solubility of Li_2CO_3 is very low (0.82 at 80°C) and the solubility of Na_2CO_3 and K_2CO_3 are high (45.1 at 80°C and 140 at 80°C , respectively). Therefore, Li^+ can be easily precipitated in the form of carbonates after adding Na_2CO_3 .

Figure R4. (a) Transmembrane rates of Li^+ , K^+ , Na^+ , Ca^{2+} and Mg^{2+} when evaporating each chloride salt solution with a concentration of 10 mmol L^{-1} . (b) The selectivity of Li^+/K^+ , Li^+/Na^+ , $\text{Li}^+/\text{Ca}^{2+}$, $\text{Li}^+/\text{Mg}^{2+}$.

General comments:

(1) The abstract exceeds the recommended words outlined in the guide for authors (approximately 220 words).

Response: The abstract has been shortened to 200 words in the revised manuscript. We copied here for reviewer's convenience:

“The demand for lithium extraction from salt-lake brines is increasing to address the lithium supply shortage. Nanofiltration separation technology with high $\text{Mg}^{2+}/\text{Li}^+$

separation efficiency has shown great potential for lithium extraction. However, it usually requires diluting the brine with a large quantity of freshwater and only yields Li⁺-enriched solution. Inspired by the process of selective ion uptake and salt secretion in mangroves, we report here the direct extraction of lithium chloride (LiCl) powder from salt-lake brines by utilizing the synergistic effect of ion separation membrane and solar-driven evaporator. The ion separation membrane-based solar evaporator is a sandwich structure consisting of an upper photothermal layer to evaporate water, a hydrophilic macroporous membrane in the middle to generate capillary pressure as the driving force for water transport, and an ultrathin ion separation membrane at the bottom to allow Li⁺ to pass through and block other multivalent ions. This process exhibits outstanding lithium extraction capability. When treating simulated salt-lake brine with salt concentration as high as 348.4 g L⁻¹, the Mg²⁺/Li⁺ ratio is reduced by 66 times (from 19.8 to 0.3). This research combines ion separation with solar-driven evaporation to directly obtain LiCl powder, providing an efficient and sustainable approach for lithium extraction.”

(2) Line 95, Fig. 1 a doesn't match with the context. It should be Figure 2. (Same for Fig. 1 b-i from line 103 to line 135)

Response: We are sorry for this mistake. The part of “Fabrication and characterization of PANI nanoarrays solar evaporator (Page 5-6)” is the description of Figure 2. We have revised them in the revised manuscript.

(3) Use consistent terminology for nanofiber. In line 99, change nano-fibers to nanofibers.

Response: “Nano-fibers” has been revised to “nanofibers” in the revised manuscript.

4. It hasn't been noted why PANI and PES are chosen.

Response: Thanks for your suggestion. To clarify this point, the corresponding description has been added to the revised manuscript (page 5, line 1-6). We copied here for your convenience:

“To realize a high photothermal conversion efficiency, vertically-aligned polyaniline (PANI) nanofiber array was selected as the upper photothermal layer owing to its excellent sunlight absorption capacity. In the middle, a hydrophilic poly(ether sulfone)

(PES) macroporous membrane with continuous water-conducting channels generates capillary pressures serving as the driving force for water transport.”

5. The introduction requires revision in some parts as it doesn't adequately introduce the main question that the study aims to address. Also, it lacks a literature review and does not showcase the novelty of this research, which is crucial to highlight the significance of the study.

Response: Thanks for the constructive suggestions, we have revised the Introduction and added a corresponding literature review to showcase the novelty of this research. We copied here for your convenience:

“Compared to conventional ion separation technologies such as ion exchange¹², electrodialysis¹³, and solvent extraction¹⁴, nanofiltration (NF) membrane separation¹⁵⁻¹⁷ is considered one of the most efficient methods for extracting lithium from brine.”
(page 3, line 10-13)

“Despite significant advancements in the separation of Mg²⁺ and Li⁺ using NF membrane processes, highly concentrated brines generate a high osmotic pressure difference across the NF membrane, requiring high applied pressure that may exceed the mechanical strength of the membrane module.” (page 3, line 18-21)

“Moreover, NF membrane separation processes solely yield Li⁺-enriched solutions, necessitating further concentration to obtain solid lithium products, which requires energy-intensive processes like thermal distillation or high-pressure RO to obtain the Li⁺-enriched solution with the required concentration. Hence, innovative material and separation process design is required to achieve more efficient and cost-effective Mg/Li separation.” (page 3, line 25 and page 4, line 1-4)

12. Liu, G., Zhao, Z. Ghahreman, A. Novel approaches for lithium extraction from salt-lake brines: a review. *Hydrometallurgy* **187**, 81-100 (2019).
13. Zhao, X., Yang, H., Wang, Y. & Sha, Z. Review on the electrochemical extraction of lithium from seawater/brine. *J. Electroanal. Chem.* **850**, 113389 (2019)
14. Sun, Y., Wang, Q., Wang, Y., Yun, R. Xiang, X. Recent advances in magnesium/lithium separation and lithium extraction technologies from salt lake brine. *Sep. Purif. Technol.* **256**, 117807 (2021).

6. It's not common to indicate the results in the introduction, i.e., line 88: with a purity as high as 94.5%.

Response: The sentence “As water continually evaporates, solid LiCl powders can be collected directly on the surface of the evaporator, with a purity as high as 94.5%.” in the original manuscript has been revised to “As water continually evaporates, solid LiCl powders can be collected directly on the surface of the evaporator.” (page 5, line 9-11)

7. What is the reference for the method of Fabrication of PANI nanoarrays evaporator? For example, why are 93 mg aniline and 114 mg ammonium persulfate chosen to fabricate the nanoarrays?

Response: The fabrication of PANI nanoarrays refers to previous studies (*Angew. Chem. Int. Ed.* **2004**, *43*, 5817 and *Adv. Mater.* **2005**, *17*, 1679), which demonstrated the polymerization of aniline in a dilute solution is a homogeneous nucleation process rather than a heterogeneous one and the polyaniline tends to grow in the form of nanofibers. The concentrations of aniline and ammonium persulfate are the optimized results referring to previous studies. The corresponding references have been cited as Ref. 51 and 52 in the revised manuscript.

51. Huang, J., Kaner, R. B. Nanofiber formation in the chemical polymerization of aniline: a mechanistic study *Angew. Chem., Int. Ed.* **43**, 5817-5821 (2004).

52. Chiou, N. R., Epstein, A. J. Polyaniline nanofibers prepared by dilute polymerization *Adv. Mater.* **17**, 1679-1683 (2005).

8. Refer to the previous comment for the fabrication of the PA membrane.

Response: The fabrication of PA membrane refers to our previous studies (Small, **2016**, *12*, 5034; *Nat. Commun.* **2020**, *11*, 2015). These references have been cited in the original manuscript. To clarify this point, the sentence “The polyamide membrane was prepared at 25.0 ± 0.5 °C and 60 ± 5 % relative humidity” has been revised to “The polyamide membrane was prepared at 25.0 ± 0.5 °C and 60 ± 5 % relative humidity following previous study^{22,53}.” in the revised manuscript (page 14, line 17).

9. In line 328, what do you mean by the concentration of LiCl, MgCl₂ feed solution were 2000 mg L⁻¹, respectively? Is the concentration of both LiCl and MgCl₂ equivalent to 2000 mg L⁻¹? In the results, their concentration has been reported as 1 mg L⁻¹ for rejection tests.

Response: Sorry for the confusion, we have revised the sentence to “Either 1.0 g/L LiCl solution or 1.0 g/L MgCl₂ solution was used as feed solution.” in the revised

manuscript (page 15, line 16-17). The salt concentration, 1.0 g/L, is consistent with the result in the original manuscript.

10. It's not apparent in the method that the membrane performance has been tested in the mixed solution. All the procedures (which are explained in the results in the current format) must be moved to their related sections in the method.

Response: The membrane performances were tested in the mixed solution under solar irradiation. To clarify this point, the sentence “A mixed solution containing a certain amount of LiCl and MgCl₂ was added in a container.” in the original manuscript has been revised to “A LiCl and MgCl₂ mixed solution was added in a container. When evaluating the effect of the Mg²⁺/Li⁺ ratio on solar-driven lithium extraction, the LiCl concentration was kept constant at 0.45 g L⁻¹ in the mixed salt solution while the MgCl₂ concentration was gradually increased from 0.45 to 4.50 g L⁻¹. When investigating the impact of salt concentrations, the overall salt concentration was increased from 10 to 80 g/L while keeping a MgCl₂/LiCl mass ratio of 1.0.” in the revised manuscript (page 16, line 1-6).

The method for Lithium extraction from simulated salt-lake brine was added to the revised manuscript (page 16, line 21-25 and page 17, line 1-9). We copied here for your convenience:

“Lithium extraction from simulated salt-lake brine. The brine was prepared based on the typical geothermal brine composition at Uyuni salar brine^{58,59}, in Bolivia, containing 268.0 g L⁻¹ NaCl, 5.1 g L⁻¹ LiCl, 66.1 g L⁻¹ MgCl₂, and 9.2 g L⁻¹ CaCl₂. The simulated salt-lake brine was used as the feed solution for the solar evaporator. Under continuous irradiation, salt crystals formed on the PANI layer surface and were collected once the surface was fully covered. 5 g of collected salt powders were added in 5 mL water to create a saturated solution. The pH value was adjusted to 11 using a saturated NaOH solution to remove any remaining Mg²⁺. The white precipitate was removed by centrifuge at 5000 rpm for 10 mins. Next, a saturated sodium oxalate solution was slowly added to the upper clear solution in order to remove any remaining small amount of Ca²⁺. The mixture was centrifuged again at 5000 rpm for 10 mins, and the upper clear solution was collected. Finally, a saturated Na₂CO₃ solution was added to the collected solution at 80 °C. The white precipitates were collected after centrifuge and dried in an air-forced oven at 60 °C for 12 hours.”

11. It had to be mentioned how the required energy for stripping water from the hydration shell of Mg and Li is calculated. (Lines 195-197)

Response: We would like to clarify that the energy required to strip water from the hydration shell of Mg and Li was not calculated by us. It refers to a previous study reported by Nightingale, E. R., Jr. in 1959, where he proposed a phenomenological theory of ion solvation and effective radii of hydrated ions (*J. Phys. Chem.* **1959**, *63*, 1381-1387). We have cited this literature in our original manuscript as Ref. 54.

12. Use relevant references on Donnan equilibrium to support the discussion made on the increase in LiCl crystallization rate from 37.3 to 61.8 g m⁻² h⁻¹.

Response: The corresponding references have been cited in the revised manuscript. We copied here for your convenience:

56. Levenstein, R., Hasson, D., Semiat, R. Utilization of the Donnan effect for improving electrolyte separation with nanofiltration membranes. *J. Membr. Sci.* **116**, 77-92 (1996).

57. Vezzani, D., Bandini, S. Donnan equilibrium and dielectric exclusion for characterization of nanofiltration membranes. *Desalination*, **149**, 477-483 (2002).

13. There is no discussion in the context of paper for Fig. 5a.

Response: The sentences “To investigate the effect of the Mg²⁺/Li⁺ ratio on solar-driven lithium extraction, the LiCl concentration was kept constant at 0.45 g L⁻¹ and the MgCl₂ concentration gradually increased from 0.45 to 4.50 g L⁻¹. increasing the LiCl crystallization rate.” are the discussion for Fig. 5a (page 10, line 15-20).

14. How did you find the concentrations in the Uyuni salar brine? Why didn't you add KCl to the simulated brine? It doesn't have any reference.

Response: The salt concentrations in the Uyuni salar brine can be found in the literature (*J. Membr. Sci.* **2021**, *635*, 119441; *Hydrometallurgy* **2012**, *117-118*, 64; *Chem. Geol.* **1980**, *30*, 57). We noted that Li⁺, Na⁺, and K⁺ have similar transmembrane rates. In the stage of lithium carbonate, effective monovalent ions separation can be achieved. Therefore, we didn't add KCl to the simulated brine.

The references about the salt compositions of the Uyuni salar brine have been added to the revised manuscript. We copied here for your convenience:

58. Rettig, S. L., B. F. Jones, F. Risacher. Geochemical evolution of brines in the Salar of Uyuni, Bolivia." *Chemical Geology* **30**, 57-79 (1980)
59. An, J. W., Kang, D. J., Tran, K. T., Kim, M. J., Lim, T., Tran, T. Recovery of lithium from Uyuni salar brine. *Hydrometallurgy*, **117**, 64-70 (2012).

REVIEWERS' COMMENTS

Reviewer #1 (Remarks to the Author):

Authors have responded to reviewer comments adequately. Therefore, there is no additional comments on this work.

Reviewer #2 (Remarks to the Author):

All my comments have been addressed, and the paper has significantly improved. I recommend it for publication.